# The Covenant Renewal Ceremony as the Main Function of Qumran

## Daniel Vainstub

Department of Bible, Archaeology and Ancient Near East, Ben-Gurion University, Beer Sheva 8410501, Israel; vainstub@bgu.ac.il

**Abstract:** Unlike any other group or philosophy in ancient Judaism, the *yaḥad* sect obliged all members of the sect to leave their places of residence all over the country and gather in the sect's central site to participate in a special annual ceremony of renewal of the covenant between God and each of the members. The increase of the communities that composed the sect and their spread over the entire country during the first century BCE required the development of the appropriate infrastructure for hosting this annual gathering at Qumran. Consequently, the hosting of the gathering became the main function of the site, and the southern esplanade with the buildings surrounding it became the epicenter of the site.

**Keywords:** Qumran; Damascus Document; scrolls; mikveh





## 1. Introduction

The subject of this paper is the yearly gathering during the festival of Shavuot of all members of the communities that composed the *yaḥad* sect.[1] After close examination of the evidence for this annual gathering in the sect's writings and analysis of the archaeological data on the development of the site of Qumran, it became evident that in the generation following that of the site's founders, the holding of the annual gathering became the main *raison d'être* of the site and the factor that dictated its architectural development.

## 2. The Covenant Renewal Ceremony

The Qumranic writings acquaint us with a most significant event in the life of the sect, a ceremony of renewal of the covenant between God and each member of the sect, described literally as "passing the covenant", which was held "year by year for as long as the dominion of Belial endures" (1QS II, 19). The celebration was an exclusively Qumranic event,[2] distinguishing the communities and their writings found in the Qumran caves from any other Jewish group or philosophy known from this period.

The most commonly accepted biblical source of inspiration for the ceremony of "passing the covenant" is the instructions for the covenant ceremony to be performed on Mounts Gerizim and Ebal in Deut 27–28 and the subsequent call to observe this covenant in Deut 29, as well as the description of the ceremony itself in Josh 8:30–35. The parallelism between the ceremony on Mounts Gerizim and Ebal and the ceremony at Qumran is not limited to the function of the priests and the Levites—blessing and cursing the people, respectively—but even extends to the vocabulary; the verb עבר "pass" with a direct object meaning "covenant," "law" or "order" normally means "to transgress" in biblical Hebrew (e.g., Deut 17:2; 26:13; Josh 7:11, 15; 23:16; Judg 2:20; 1 Sam 15:24; 2 Kgs 18:12; Isa 24:5; Hos 6:7; 8:1; Ps 148:6; Esth 3:3; Dan 9:11), as well as in Qumranic Hebrew (e.g., 1QM LV, 7, 17 (לעבור בריתי); 1QS V, 7, 14; CD I, 20). The only occurrence in the Bible of עבר and ברית using the preposition ב, literally "to pass in a covenant", is in Deut 29:11, while its only occurrence in the Qumran scrolls is in the description of the renewal of the covenant inspired by the passage in Deuteronomy, both occurrences expressing the passing of the participants

in the ceremony[3] and differing from Rabbinical Hebrew, in which the same expression means "to transgress", just as without the preposition בּ (e.g., m. Ḥal. 1:2; 5:4; m. Pesaḥ. 3:1; m. B. Meṣ. 5:12). Another biblical source that could have influenced the Qumranic terminology for the ceremony is Ezek 20:36–38, where God says he will pass (וְהַעֲבַרְתִּי) the selected people under his shepherd's staff into the covenant (הַבְּרִית), separating out (וּבָרוֹתִי) the wicked ones.[4] In addition, the term "blesses himself in his heart" (יתברך בלבבו) and its meaning in this context are taken from Deut 29:19 (see below).

The annual celebration is recorded in both the Community Rule and the Damascus Document[5] and is one of the points that they have in common (Hempel 2010, especially p. 128), a fact leading to the conclusion that the event was a meaningful one in the life of the sect, in which all full members of the different types of communities described in these two compositions converged on one place once a year.

In the Community Rule, the description of the celebration is located in a very honored place at its beginning after the introduction (1QS II, 19–III, 12).[6] Some manuscripts of the Community Rule coming from Cave 4, however, lack these passages.[7]

The section of the Damascus Document that explicitly discusses the covenant renewal ceremony is missing in Cairo Genizah's versions of the composition, but it is partially preserved in three (4Q266 1, 4Q270 7 II and 4Q269 16, 15–18; see Hempel 2009, p. 371) of the ten manuscripts of the composition found in the Qumran caves (4Q266–273; 5Q12; 6Q15). Most probably, the section was placed at the end of the composition (Baumgarten et al. 2006, pp. 3–4; Hempel 2010, p. 127), and in Fraade's opinion, the entire document was "an anthology that was drawn upon so as to provide performative 'scripts' . . . for the annual covenant-renewal ceremony" (Fraade 2011, pp. 238, 245), a view shared by Falk (Falk 1998, pp. 228, 234). This conclusion is in line with that of Mandel, who concluded previously that the Damascus Document was built in inclusio (Mandel 2004).

One of the most prominent and well-known differences between the Damascus Document and the Community Rule is that the Damascus Document regulates, among others, communities catalogued as מחנות "camps", whose members could be married and have children, like the kind of Essenes recorded by Josephus in *J.W.* 2.8.8. Undoubtedly, the annual convergence of community members who not only came from different places but also followed different rules and social codes was a very significant event that required a logistical operation on a large scale.

Other texts connected with the covenant renewal ceremony are found in the following Qumranic compositions: 4QBerakhot (4Q286–290), which contains a text that was declaimed in the covenant renewal ceremony (Nitzan 2000; Nitzan 2014, pp. 179–85; Falk 1998, p. 236); Scroll 5Q13, which has survived very partially[8] and, most probably, Scroll 4Q275. The composition 4QBerakhot contains benedictions to God and curses on Belial and all those who "belong to his destiny" (meaning those who do not belong to the *yaḥad* sect), declaimed by the participants of the ceremony based on mystical Merkabah speculations (Nitzan 2000). The exalted words of the benedictions, which were most probably pronounced at one of the climaxes of the ceremony, were intended to reinforce the participants' belief in the unique perceptual foundations of the *yaḥad* sect with regard to the true calendar and festival dates, the exclusive legitimacy of the priests of the *yaḥad* and the selection by God of the Sons of Light (members of the *yaḥad* participating in the ceremony), while God Himself and His angels themselves take part in the ceremony while standing in their heavenly sanctuary (Nitzan 2014, pp. 180–82, 85). Undoubtedly, the ceremony was a profound spiritual experience that endowed the participants with a strengthened faith that would accompany them for an entire year after their return to their communities. Scroll 4Q275, of which only three small fragments survive, mentions people "walking the paths . . . in the third month (Sivan)" (י]הולכים אֹת שבילי ה̇, [ ] בחודש השליש[ן).[9]

The annual covenant renewal ceremony is the outcome and the sequel of a fundamental conception in the ideology of the sect: the covenant between God and His chosen Sons of Light that separates them from the Sons of Darkness and is the continuation of a chain of covenants made by God, such as the Revelation at Mount Sinai, which began

with Noah after the Deluge. Through these covenants, God elected His chosen ones by the principle of reduction through the generations, and the members of the *yaḥad* were the Sons of Light chosen by God at that time (Nitzan 2014, pp. 170–74; Schiffman 2004), as stated at the beginning of the Damascus Document: "but when He remembered the covenant of the forefathers, He left a remnant to Israel and did not allow them to be totally destroyed" (ובזכרו ברית ראשונים השאיר שארית לישראל ולא נתנם לכלה). However, unlike the preceding covenants, the present one, called the "new covenant" in the Damascus Document, has a very prominent personal dimension: the man who is accepted as a member and joins the sect "comes into the covenant" (בא בברית) and is thereby incorporated into the covenant between God and the *yaḥad* sect. His participation for the first time in the annual covenant renewal ceremony terminates his period of candidacy, after which he is finally accepted as a full member of the sect (see Falk 1998, p. 219; VanderKam 2009, pp. 226–27). That is to say, each year, together with the renewal of the covenant, new members "come into the covenant" after they have successfully completed their candidacy in one of the communities of the *yaḥad*. All members of the sect were obliged year by year to "pass the covenant" (שנה בשנה כול יומי ממשלת בליעל), which is the special covenant between the *yaḥad* communities and God, and participate in the annual covenant renewal ceremony until the end of the "rule of Belial," namely until the end of days. Non-compliance with that condition would lead to a member's being expelled from the sect, the pronouncing of divine curses on him and his inclusion among the Sons of Darkness. This unique annual gathering, in which each member ratified his belonging to the sect, would have brought the members of the *yaḥad* to elation on the one hand and holy fear on the other.

Beyond the fact that holding the rite of the renewal of the covenant was a halakhic obligation of each member of the sect, the annual gathering of all members of the congregations of the *yaḥad* in one place was probably the most important event of the year. The gathering was also attended by the sages and leaders of the sect, including all the holders of the post of מבקר,[10] who headed the different communities in the country. The shared prayers and rituals conducted during the communal Shabbat (below), and the festival only added to the feelings of transcendence and exaltation. In Fraade's opinion, the annual gathering was the most important ritual and liturgical event in the annual festival cycle of the Qumran community.[11]

As detailed in the Community Rule, the ceremony was conducted similarly—but not identically—to the covenant renewal ceremony on Mounts Ebal and Gerizim in the days of Joshua, as described in Deut 27–28 and Josh 8:30–35. All the members of the *yaḥad* gathered in one place and walked in line between the priests and the Levites. The priests solemnly read the benedictions, the Levites read the curses and the members of the *yaḥad* expressed their consent by saying "Amen, amen!" The sincere belief that God and His angels saw the true intentions and thoughts of every sect member who was about to participate in the ceremony and that a parallel ceremony was being held in heaven with the participation of angels (Nitzan 2014, pp. 179–85) would create fear and trembling in the heart of a participant who feared that there might be a defect in the purity of his faith and that he might sin by swearing a false oath in the sight of God. A person who participated in the ceremony even though his faith was incomplete "blesses himself in his heart" (יתברך בלבבו) and was damned, as "his spirit shall be destroyed without pardon", "all the curses of this covenant shall cling to him, and God will set him apart for evil. He shall be cut off from the midst of all the Sons of Light" (1QS II, 11–18), curses that would terrify and deter any member who was not completely sure of the purity of his thoughts. A member of the *yaḥad* who refused to participate in the ceremony was considered one who "refuses to enter the covenant of God" (מואס לבוא בברית אל), was immediately expelled from the sect and his sentence was decreed: "unclean! unclean! shall he be" all the days of his life, "he shall neither be purified by atonement, nor cleansed by purifying waters, nor sanctified by seas and rivers . . . " (1QS II, 26–III, 6). This description is consistent with Josephus' description (*J.W.* 2.8.8) of the Essenes who were expelled from their communities and wandered the land, miserable and humiliated.

The description of the ceremony in the Damascus Document includes a sentence containing two pieces of information that are of critical importance for our purposes and are missing in the Community Rule: "And all [the inhabitants] of the camps (המחנות) shall assemble in the third month and curse anyone who deviates either to the right [or to the left from the] Torah."[12]

A. From this description, we learn only that the ceremony took place in the month of Sivan. Still, following the accepted opinion, the reference is to Shavuot (Weinfeld 1990, pp. 20–21, 23, 30–31; Nitzan 2014, pp. 172–74), a fact that is not surprising, since according to a very common belief in the Second Temple period, the previous covenants, including the covenant at Mount Sinai and the reception of the Torah, were established on the same date (Nitzan 2014, pp. 172–73). The Book of Jubilees (6:11, 17) also seems to allude to the holding of the renewal of the covenant ceremony on Shavuot: "you should make a covenant with the children of Israel in this month . . . they should celebrate the Shavuot feast in this month once a year, to renew the covenant every year" (כי יעשו את חג השבועות בחודש הזה פעם בשנה לחדש את הברית בכל שנה ושנה) (see Weinfeld 1990, p. 30; Licht 1965, p. 56; Nitzan 2014, pp. 172–73; Milik 1957, p. 77). In Wacholder's opinion, the ceremony was held on the 16 of Sivan, the day following Shavuot (Wacholder 2007, p. 367). According to Baumgarten, it was held not on the day of Shavuot itself but during the three days preceding it,[13] called the "three days of restriction" in the Pharisaic tradition based on the days of communal purification before the Sinaitic covenant (Exod 19:10–12). Hence, in this interpretation, the gathering with all its rituals began in the week preceding Shavuot. In the Qumranic calendar, Shavuot always falls on a Sunday, so in fact it was a two-day gathering including the festival and the Shabbat preceding it.

B. The second piece of information is that all the *yaḥad* congregations of the camp type were obliged to gather for the ceremony in a single place, a law that seemingly also applied to the *yaḥad* communities that lived in the cities beside the general Jewish population, according to two partially preserved fragments of the Damascus Document: (1) "the hol[y in their camp]s [and] their cities in al[l" (4Q266 5 II) (4Q266 5 II; Baumgarten et al. 2006, p. 36) and (2) "for all who dwell in their [c]amps and all who d[well in] their [towns.] Behold, it is all w[ritten] in accordance with the final interpretation of [the] Torah (4Q270 7 II, 13)."[14] This critical evidence is situated shortly after a mention of the countrywide gathering in the month of Sivan in the following order: Line 11 " . . . in the] third month and cur[se . . . " . . . Line 12 " . . . this is the interpretation of the precepts" (פרוש המשפטים), most probably an introductory phrase before some details of the gathering mentioned above.

The camps mentioned in the Damascus Document are the isolated communities of the *yaḥad* sect scattered throughout the country, and it seems that the directives in the Community Rule with regard to small communities of fifteen, ten or even fewer members "in all their residences" also refer to them (1QS VI, 2–5; VIII, 1–2).

From all of the foregoing, it appears that once a year, before Shavuot according to the Qumranic calendar, all the members of the congregations that comprised the *yaḥad* sect gathered together from all over the country in the central place of the sect, and the obligatory annual covenant renewal ceremony took place only here. This mass migration of members of the camps undoubtedly added to the uniqueness of the event and intensified the emotion of its participants, since Shavuot is one of the three Pilgrimage Festivals, when Jews were required to make pilgrimage to the Temple in Jerusalem with their first fruits. Thus, the singular ceremony of the renewal of the covenant was, in fact, an alternative to the pilgrimage to the Temple and a challenge to the customary calendar.

The practical significance of the above is that once a year, many hundreds, or perhaps even several thousands of people converged on the central place of the sect and needed to be accommodated for a few days or even a week. Members of the camps and the urban communities had to arrive in time to prepare for the Shabbat and the Shavuot festival with the covenant renewal ceremony that followed. It is very likely that the gathering was also made use of for study, reflection and discussion in the days leading up to the event (Schiffman 1989, p. 13; Fraade 2011, pp. 238–39), and scrolls probably found their way to

and from the main site and the communities scattered across the country.[15] The Shabbat before Shavuot is the eleventh Shabbat according to the counting of the sect. The services held on this day have been preserved, albeit in a very fragmented way (Alexander 2006, pp. 38–40; Mizrahi 2019, pp. 5–35).

### 3. The Annual Countrywide Gathering of All the Yaḥad Communities

Since Shavuot was the only time of year during which there was a halakhic obligation for all members of the sect to gather, textual evidence for an annual countrywide convention that does not explicitly mention the covenant renewal ceremony should also be linked to this date (see Licht 1965, pp. 14–17). According to a new interpretation that I will present below, the rules regulating the annual gathering of all congregations for the covenant renewal ceremony are also included in the Damascus Document. In my opinion, the annual countrywide gathering was called by the members of the sect "the assembly of all the camps" (מושב כל המחנות).

Pages 12–14 of the Damascus Document contain a series of rules (*serekhs*) (Wacholder 2007, p. 344) that are commonly read in succession and interpreted as a general description of the sect. However, in my opinion, we see here two orderly rules for the two well-differentiated types of communities that made up the sect (urban communities and communities living in separate settlements) and a third rule regulating the annual countrywide gathering (see Table 1). As will be discussed in more detail in due course, these regulations were probably formulated at the end of the Hasmonean period or the beginning of the Herodian period, when various arrangements were established for the different *yaḥad* communities scattered throughout the country.

The word מושב in this cluster of rules is sometimes interpreted as "habitation" or "settlement", its common meaning in the Bible and in many Qumranic scrolls, especially in texts recalling or resembling biblical ones. However, this meaning is inadequate in the context of the present cluster of rules, where the meaning "assembly" fits better.[16] The word מושב in the meaning of "assembly" is common in Qumranic Hebrew in legal or regulatory texts, as in the well-known expression מושב הרבים that appears in the Community Rule in the context of the regulations for the assembly of the רבים.[17] Moreover, the expression סרך מושב הרבים in the Community Rule clearly means "the rule of the assembly of the רבים", and for this reason, the word מושב does not occur in the fourth rule of the Damascus Document, since this rule does not deal with behavior in the assembly of the רבים as in the Community Rule, but rather with obligations not related to their meetings. Likewise, in the regulations of the Community Rule, the term מושב is used in the meaning of "assembly" to express exclusion from the assembly for a number of meetings as punishment of a member.

**Table 1.** Rules in Damascus Document.

| Rule | Damascus Document | | |
|------|-------------------|------|------|
| 1 | XII, 19–22 | סרך מושב ערי ישראל | Rule of the assembly of (the communities living in) the cities of Israel |
| 2 | XII, 22–XIV, 2 | סרך מושב המחנות | Rule of the assembly of the camps |
| 2a | XIII, 7–19 | סרך המבקר למחנה | Rule of the "מבקר of a camp" |
| 3 | XIV, 3–12 | סרך מושב כל המחנות | Rule of the assembly of all the camps |
| 3a | XIV, 8–11 | (סרך?) המבקר אשר לכל המחנות | (Rule?) of the "מבקר of all the camps" |
| 4 | XIV, 12–17 | סרך הרבים | Rule of the רבים |

The first rule reported is the סרך מושב ערי ישראל (XII, 19), namely the rule for the assembly of every community living in one of the cities of the country alongside the general Jewish population, similar to those described by Josephus in *J.W.* 2.8.4 ("They occupy no one city, but settle in large numbers in every town") and Philo of Alexandria (Eusebius of

Caesarea, *Praep. ev.* 8.11.1) ("They dwell in many cities of Judaea and many villages, and in large and populous societies.").

Next comes the סרך מושב המחנות (XII, 22–23), namely the rule for the assembly of each of the isolated communities with at least ten members (CD XIII, 1) and for the leader of each community who bears the title מבקר אשר במחנה or מבקר אשר למחנה, מבקר למחנה, undoubtedly to be identified with the "curator" or "overseer" (ἐπιμελητής, ἐπίτροπος) who, according to Josephus (*J.W.* 2.8.6), headed every Essene community. This long rule, which includes the סרך המבקר למחנה, or "rule of the מבקר of a camp", ends with the closing sentence מושב המחנות וזה ("and this is the assembly of (each of) the camps"), which begins in line XIII, 20 and ends at the end of the second line on page XIV.

Immediately after that, from the third line to the twelfth line[18] on the same page comes the סרך מושב כל המחנות ("rule of the assembly of all the camps").[19] This is clearly a joint session of members of all camps in the country, since the directives for meetings of the members of each camp separately are given in the previous rule. In my opinion, it refers to the general annual gathering of all the *yaḥad* communities held at Qumran during Shavuot, which is the only annual countrywide convention of the sect known from the written sources. At the head of this assembly was a person who held the title of מבקר אשר לכל המחנות, the "מבקר of all the camps" (XIV, 8–9), and was in charge of all matters of the assembly. All the participants in the gathering were registered by name in a careful record while establishing a hierarchical order between them; בשמותיהם איש אחר אחיהו יפקדו כלם בשמותיהם ... ויכתבו seems to refer to the hierarchical location of each member in the all-*yaḥad* convention, unlike his hierarchical position (*tikkun*) within the community to which he belongs. For the purpose of determining this hierarchy, the priests, the Levites, the common Jews and the proselytes of each and every community were separated in the general assembly of all the communities. The hierarchical location of each individual within one of these four countrywide groups was determined in a very complex crossword puzzle. The understanding proposed here—that the third rule differs from the second rule that regulates the rules in each camp separately—is required for the following reasons:

A. The explicit term "assembly of *all* the camps" differs from "assembly of the camps" in the previous rule.

B. The title of the person in charge of the assembly is, as aforesaid, "מבקר of *all* the camps" and not "מבקר of the camp", as in the previous rule.

C. The requirements for appointment of a person as מבקר of a single camp are specified in the "rule of the מבקר of the camp" contained within the second rule. These requirements are general education and leadership. On the other hand, in order to fulfill the role of "מבקר of all (שלכול [המחנות] in 4Q266 10 I) the camps" (XIV, 8–12), there is an age requirement ("from thirty years to fifty years old"). In addition, his education and knowledge must be outstanding, and he must be "accomplished in every mystery (revealed to) men"; that is, he must be versed in all the secret information given to the sect and "in every tongue as (spoken) by their families".[20] In other words, he had to be familiar with the various languages spoken by Jews in the land at that time—Hebrew, Aramaic and Greek—in order to communicate with members of the various communities who came from all over the country (Baumgarten and Schwartz 1994, pp. 56–57; Wacholder 2007, p. 98, note 278, 350). The priest who presides over the general gathering is subject to an age restriction ("from thirty years to sixty years") and must be "an expert in the book of Haguy and in all the injunctions of the Torah to interpret them correctly" (XIV, 6–8). In contrast, in the second rule, there is no age restriction for the priest of a single camp, and he is required only to be "versed (מבונן) in the Book of Haguy". The rule also states that "if he is not competent in all these matters", that is, if he is not learned or even "foolish" (פתי), meaning ignorant, one of the Levites or the "מבקר of the camp" should make decisions in his place (XIII, 2–6).

On this point, it is important to stress some elements of the syntax used in the text because of their crucial importance for the subject treated in this paper. As was correctly observed by Naudé and Miller-Naudé, in Qumran Hebrew—as well as in Biblical Hebrew—the expression כל + a definite plural noun expresses the totality of the (specific) group.

Thus, מושב כל המחנות can mean only the assembly of all the camps together, and in the case of והמבקר אשר לכל המחנות, in their words "And the Inspector who is over all the camps," ... "the meaning is that there is a single inspector who is over all the camps, not one inspector over each of the camps."[21] Hence, in accordance with the standard syntactical rules of Qumran Hebrew, our text undoubtedly deals with a gathering of all the communities of the sect and with a person appointed over this gathering.

D. The expression באי העדה (XIV, 10) in the third rule seems to refer to each member of the entire sect, as opposed to כל באי המחנה (XIII, 4) and בני המחנה (XIII, 13) in the second rule, referring to each member of a community. In the second rule, הנוסף לעדתו (XIII, 11) is a person who has joined the sect by joining one of its communities.

E. One of the most important tasks of the "מבקר of a camp" is to supervise the entire process of admission and absorption of the new entrants, similar to what is described in the Community Rule. In contrast, the "מבקר of all the camps" is not in charge of accepting new candidates and accompanying them through their lengthy admission process. Among his duties was "by his authority the members of the עדה shall enter, each in his turn", a sentence that can be understood in two ways. One possibility is that the באי העדה are all members of the *yaḥad* sect, each of them approaching the "מבקר of all the camps" according to his combined hierarchical location. The other possibility is that the באי העדה are those who have completed the process of candidacy in their communities this year and have "come to" the sect (see Qimron 1990, p. 117) in the framework of the covenant renewal ceremony and must now be registered according to their combined location.

The fourth rule is the "rule of the רבים". The רבים are the full members of the sect.[22] The term רב, found at Qumran only in its plural form, for a full member of the sect is derived from the meaning of the word as great or sublime in value, assets or wisdom and not in the sense of quantity (as in Carmignac 1971), exactly as in the title of sages in mainstream Jewish circles.[23] This meaning of the word is found in the Bible (e.g., Gen 25:23; Num 22:15; Isa 19:20) and in a few occurrences in Rabbinical Hebrew and the Apocrypha.[24] One might ask whether the rule refers to the רבים in each community or to the רבים of all the groups together at the annual gathering. Indeed, the location of the fourth rule immediately after the rule of the annual general gathering may tip the scales toward the second possibility, for if it describes the רבים in each camp, one would expect it to come between the second and third rules. On the other hand, the proponent of this interpretation must justify the duplication of the role of the מבקר על כל המחנות and the מבקר על הרבים, unless these are the titles of two different officials functioning at the annual gathering. This is quite possibly similar to what we read in the Community Rule, in which various officials in charge of public affairs are mentioned: האיש המבקר על and האיש הפקיד בראש הרבים, האיש המבקר על הרבים מלאכת הרבים.[25] According to the fourth rule, each member of the רבים must set aside for the community coffers the monetary equivalent of the wages of at least two working days a month "to fill all their needs" (XIV, 12) and "all the services provided by the חֶבֶר" (XIV, 16) and to fund charities and support the needy. Therefore, each member of the community contributed to the congregation about a monthly salary per year, a considerable sum that made it possible, among other things, to hold the annual general convention. This may also be an explanation for the funding of the site of Qumran and its coin hoards. The undertaking "(for) all the services provided by the חֶבֶר" (XIV, 16) is to finance all the expenses of the group ולא יכרת בית החבר מידם "so that [the house of the חֶבֶר] not perish for lack of their (support)" (CD XIV, 16–17 based on 4Q266. See Baumgarten et al. 2006, pp. 62–63; Wacholder 2007, pp. 98–99). Wacholder (2007, pp. 351) interprets the phrase בית החבר as a nickname for a special group of members of the community in charge of solving social problems, although one cannot exclude the possibility that this is the nickname of the public structure of the community in which its institutions are concentrated.[26]

As part of the covenant renewal ceremony and in addition to it, a special ceremony was held for the young joiners, the sons of the members of the camps (XV, 1–2), apparently when they reached the age of twenty (Qimron 1990, p. 116). Presumably, this ceremony was held before the covenant renewal ceremony, for it was the gateway to the sect for

these young people and prepared them for the general ceremony. The special ceremony was called שבועת הבנים "the oath of the sons" and also consisted of reciting blessings to those who joined and curses on those of them who dared to break God's laws after joining, together with the swearing of the young people to both by saying "Amen". However, the recitation of the curses in this ceremony included a very unusual practice that added to the excitement of the event: in each curse, the explicit Tetragrammaton was pronounced, rather than its substitutes Adonai or Elohim.[27] In this extraordinary act, intended to instill terror in the hearts of the young, the curses sounded as if God Himself was hurling the curses in the face of anyone who dared to deviate from the straight path. It is quite possible that the priest and the "מבקר of all the camps," who played a part in the annual convention, were chosen from all the priests and מבקרs of the various congregations to fulfill this role, unless they were the מבקר and priest of Qumran, the main site of the sect.

## 4. The Site of Qumran and the Covenant Renewal Ceremony[28]

From the extensive excavations of De Vaux at Qumran in the 1950s, as well as surveys, excavations and other studies conducted at the place, this site has emerged as a unique site whose features do not match those of any other known model or type of archaeological site (Figure 1).

This uniqueness has given rise to over a dozen different proposals for its definition in the last seventy years, from an Essene center to an agricultural farm and from a road station to a fortress (see Broshi and Eshel 2004). It seems that despite the thousands of papers and books published about Qumran and the caves surrounding it that have piled up on the shelves of scholars, the most basic controversies concerning the establishment and functioning of the site remain. In my opinion, the uniqueness of the site is precisely the reason for this; it clearly indicates its belonging to a special and exceptional human group, very different from the general population in its customs and needs expressed in the architectural and ceramic finds. Many scholars, including De Vaux himself, have rightly pointed to important correspondences between the archaeological finds at Qumran and the information we have about the Essenes from the literary sources and sectarian scrolls discovered in caves in the area. Among these are the great correspondence between the unusual multiplicity of ritual baths (*miqva'ot*) in the site and the purification customs of the Essenes (Collins 2010, p. 205), the geographical location conforming with the testimony of Pliny the Elder (*Natural History*, book 5, chapter 17) and Dio Chrysostom,[29] and the similarity of the ceramic finds of the site to those of the caves in which the scrolls were discovered. The ceramic and numismatic finds clearly link the Qumran ruins with the many dozens of natural and artificial caves and leveled surfaces in its vicinity. They are all connected by a network of paths and together formed an archaeological complex that was unique in the country during the Hellenistic and Roman periods, facts that reinforce the opinion of most scholars that the complex served as a major site for the *yaḥad* sect.

In the following, I will discuss a number of archaeological features in the southern part of the Qumran ruins that support our opinion and throw much light on it, as not all of them have been highlighted in the research.

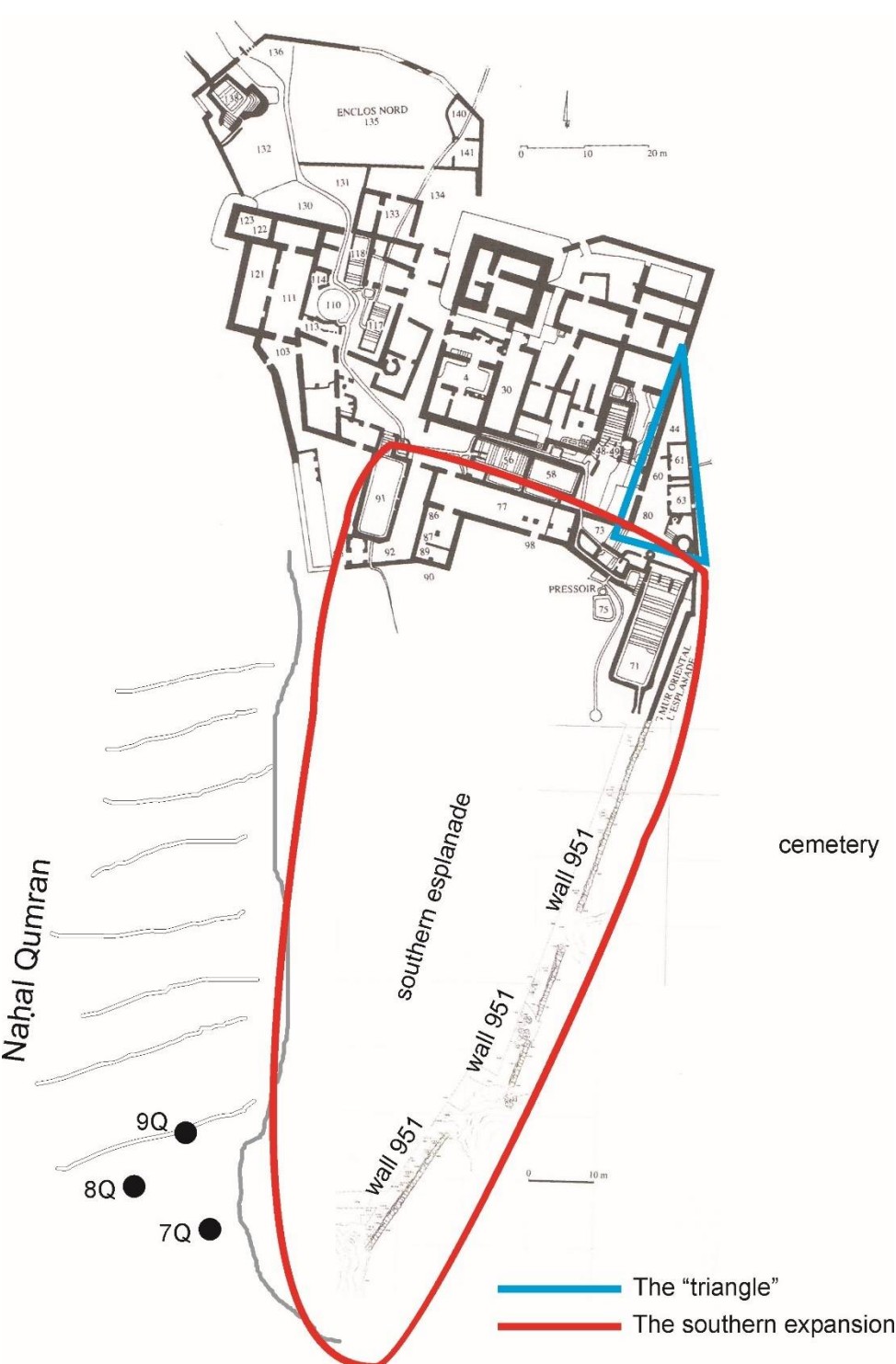

**Figure 1.** Based on Humbert 1994, p. 204 and Humbert and Chambon 2016.

### 4.1. The Assembly Hall (L77) and Miqveh 56–58

The elongated room L77 (Figures 1–3) in the south section of the site was interpreted by De Vaux as the dining and assembly hall of the sect, and his logical proposal was widely accepted.

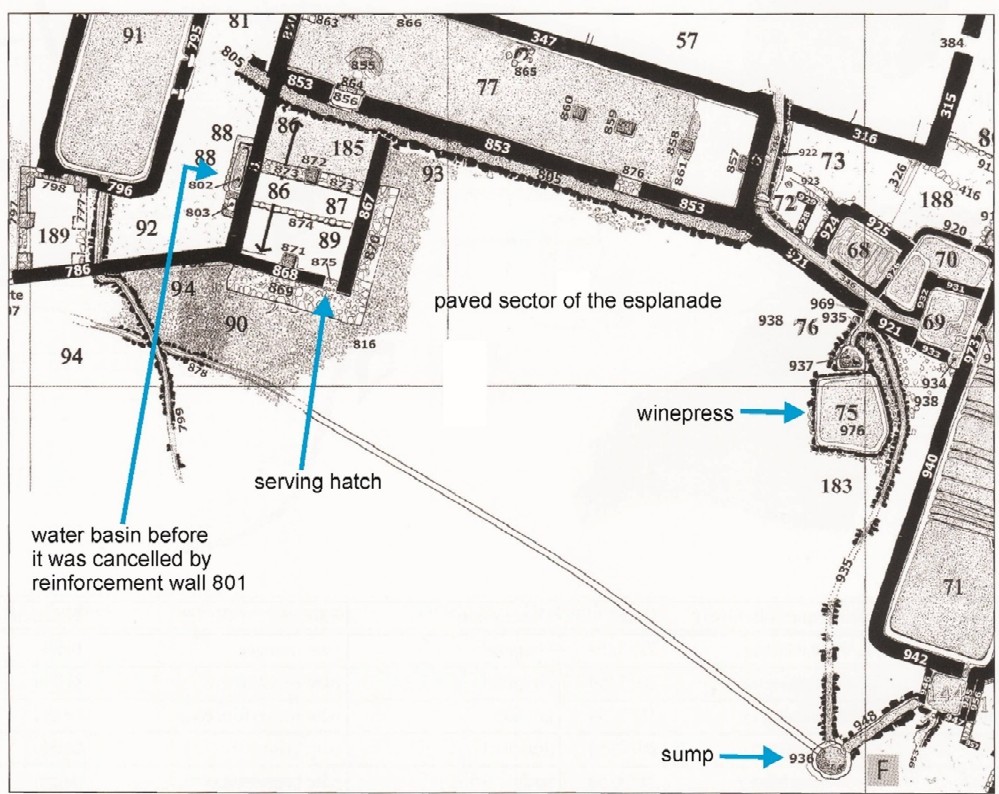

**Figure 2.** Based on Humbert and Chambon 2016.

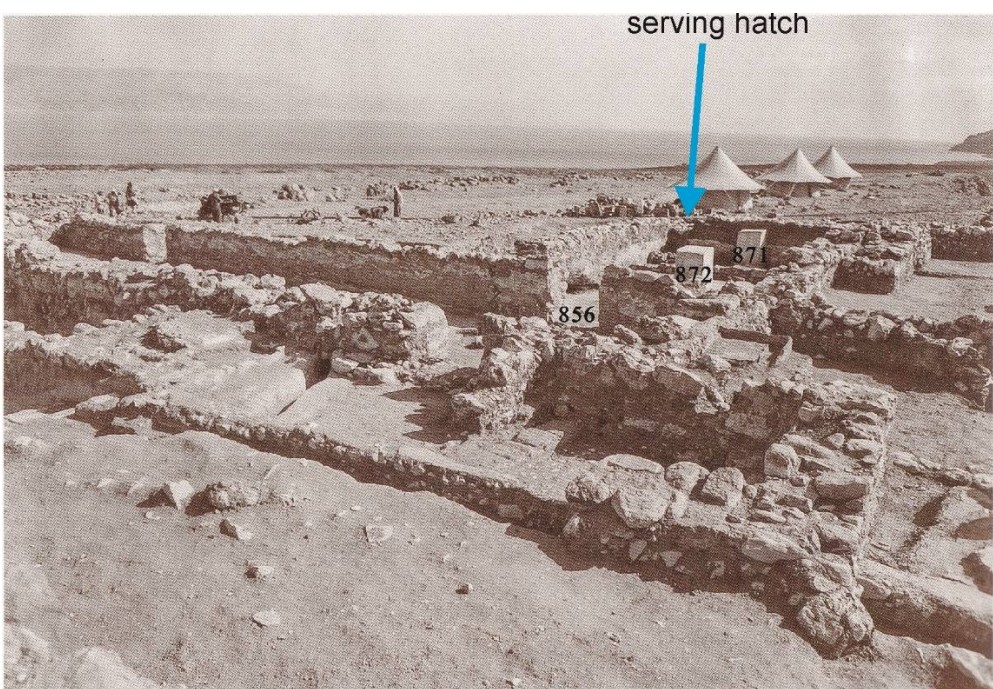

**Figure 3.** Rooms 77 and 86. Based on Humbert and Chambon 2016.

Two well-designed openings connect it to the forecourt of *miqveh* 56–58 in the north and to locus 86 in the south, where an assemblage of tableware and serving utensils was discovered (below). In this interpretation, it was in this hall that from the end of the Hasmonean period or the beginning of the Herodian period, as described in the Community Rule, the members of the רבים body of the *yaḥad* sect—that is, the full members of the sect—

held their two daily meals and study meetings after immersing in *miqveh* 56–58.[30] Locus 86 was interpreted as a pantry where the utensils needed for the meals were stored. The staircase of *miqveh* 56–58 is divided into three by two symbolic plaster partitions, unlike the customary arrangement in most of the site's *miqva'ot* and in Judea in general, in which the stairs are divided into two by one plaster partition. This triple division was later implemented in *miqveh* 71 as well (below). Hall 77, measuring 22 × 4.5 m with the opening of water drain 866 installed in its northern wall (Humbert and Chambon 2016, p. 315), could accommodate more than a hundred people. The location of the hall at the southern end of the complex, detached from the main architectural units of the site and far from the main entrance to the site, which was probably on its northern edge, also fits the proposed interpretation, since the daily study of the רבים included secrets. These secrets were revealed to members only after their full acceptance after more than two years of candidacy, and candidates were not allowed to hear them (see Bar-Ilan 1997). The secrets included the names of angels and exposure to mysteries of the cult. The terrible oath not to reveal them even under severe torture sworn by those who joined the acceptance ceremony was so characteristic of the *yaḥad* sect that it became known to many and reached the ears of Josephus and Philo of Alexandria, who noted the oath among the special features of the sect in accordance with what we now know from the Rule of the Community (Ibid.). The candidates had to immerse in a different *miqveh* and dine in another place.[31] It therefore makes sense that the *miqveh* of the רבים, their assembly room and the adjoining utensils room should be built in an isolated place at the end of the site, detached from the other building units rather than being separated merely by a wall. Nevertheless, one could easily enter the site from the north and cross it through the central north-south axis that separates the architectural units of the site. In this context, it is worth looking at the words of Josephus (*J.W.* 2.8.5) about the dining room of the Essenes:

> "and after this purification they assemble in a special section which none of the uninitiated is permitted to enter; pure now themselves, they go into the dining-room, as into a certain holy temple."

Hall 77 had another opening, 876, which connected it with the terrace known as the southern esplanade. This terrace is of great importance in the analysis of the whole site proposed in this article and will be discussed in detail below.

### 4.2. The Pantry (L86–89)

Most researchers believe that room 86 (8.20 × 5.00 m) was designed and built as a single unit with hall 77 and functionally connected to it (Figures 2–4).

This conclusion derives from the following facts:

A. As stated above, the only access to room 86 is through opening 856, located in the south side of hall 77.

B. In both of these rooms, and only in them, is a special architectural element that has no equal in the archaeology of the Land of Israel: rectangular pillars of sorts, built of bricks and plastered, that lie on the floor, although their dimensions are different in each of the two buildings. According to some researchers, such as Stacey (Stacey 2013, p. 50), Magness (Magness 2004, p. 100) and Magen and Peleg (Magen and Peleg 2018, p. 46), these were designed to support a roof or a shed, while others believe that they were intended for a different purpose.[32] According to Pfann, these pillars could have been used to temporarily place trays or other serving utensils (Pfann 2006, pp. 166–67).

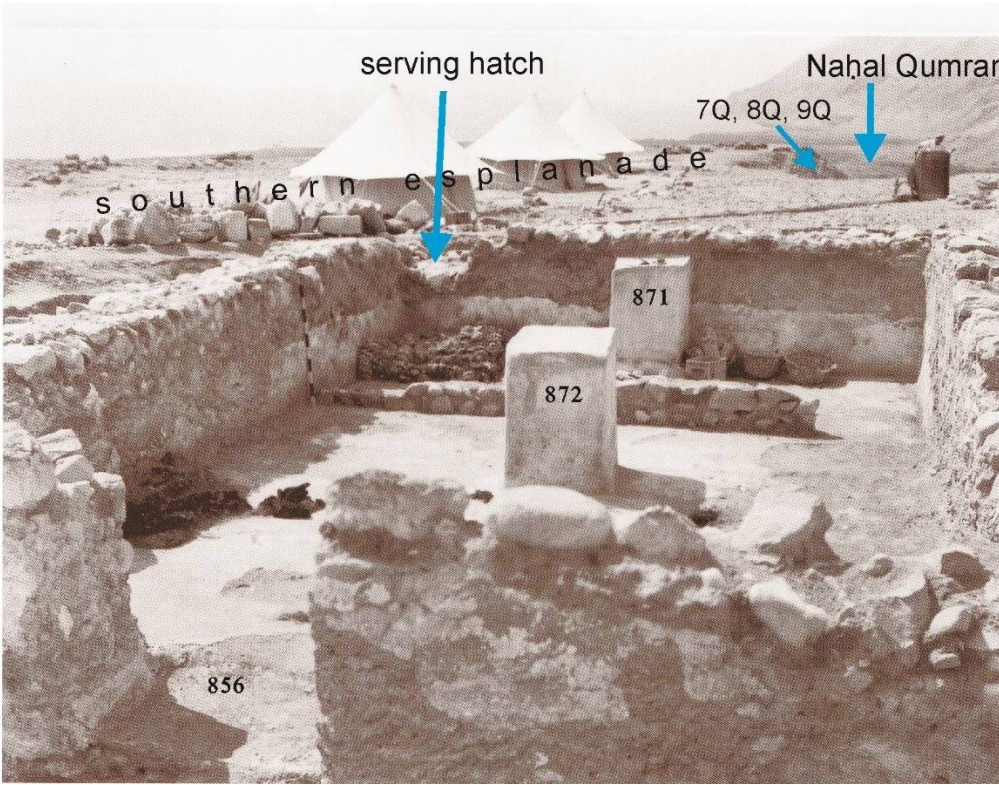

**Figure 4.** Room 86. Based on Humbert and Chambon 2016.

In room 86, there were two pillars or plastered blocks made of unbaked mud-bricks, 871 and 872, whose original height from the floor was about 90 cm. Photographs taken during De Vaux's excavations show pillar 872, which stood in the middle of the room. Although the pillar was later damaged by conservation work, the photographs clearly show its plastered top surface, which was concave so that a large vessel could be securely placed on it and the food it contained served into dishes (Humbert and Chambon 2016, p. 72). The top of pillar 871, adjacent to wall 868, was not preserved, but since the preserved parts of this pillar are clearly identical to those of pillar 872, it is likely that the two pillars were identical and that the top of pillar 871 was also concave to allow large vessels to be securely placed (Wagemakers and Taylor 2011, pp. 143–45). Since the pillar is located only 50 cm from window 875 which, in my opinion, served as a serving hatch (below), it must be assumed that the two pillars and the hatch are connected in a serving chain from room 86 to the southern esplanade. Using the two pillars with their concave surfaces, two people could simultaneously fill bowls with stew and serve them to the esplanade through the hatch. Alternatively, empty dining utensils could be passed through the window so that they could be filled in the northern part of the esplanade next to the building and distributed from there to the people seated in the esplanade. In its original design, similar to hall 77, room 86 had a regular supply of water through opening 851 in its western wall, which led to a plastered basin (L38) on the other side of the wall. When the structure became unstable and it was decided to build three reinforcement walls, reinforcement wall 801 canceled the basin.

During its existence from its construction in the first century BCE until its destruction along with the entire site by the Romans in 68 CE, room 86 underwent significant changes: reinforcement walls 801, 869 and 870 were erected around it, railing 874 was built inside it and wall 873 was built in its middle, which effectively eliminated the use of pillar 872 and the southern half of the room and left only the northern half (L185) in use. In the southern part of room 86 (L89), De Vaux discovered a very rare find: piles of beautifully arranged pottery vessels placed upside down, numbering over a thousand. It seems that the vessels had been cracked and broken due to the undermining of the structure, and the people

of Qumran decided to leave them in place and later even completely cancel the southern half of the room, decisions that leave us with a room in a kind of frozen state.[33] Among the tableware are 720 deep bowls, 209 plates and 81 cups, and among the serving vessels are 38 large, deep bowls and 21 jars.[34] The absence of cooking utensils in the assemblage has sharpened the understanding that this is not a general warehouse for the pottery on site (Pfann 2006, p. 162). It is probable that at the time of the Roman conquest of the site, there were similar piles of utensils in the northern half of room 86 (L185) and that these were taken by the occupiers for their own use. After the Roman conquest, room 86 was abandoned in all its parts, and its remains remained outside the compound.

One of the problematic issues in the understanding of the site in general and room 86 in particular is the disproportional amount of pottery found in it (Mizzi 2017a, p. 12), primarily bowls, plates and cups which were mostly plain, undecorated, modest and practical, in line with the ideology of the sect (Crawford 2019, pp. 196–99). Why would the dozens of permanent residents of Qumran need thousands of dining vessels? This disproportion was one of the reasons behind the proposal of Magen and Peleg to interpret the site as a pottery production center. Their theory, however, is refuted by INAA and petrographic analyses of the Qumran pottery carried out in recent years, which show that part of this pottery was imported to Qumran from other areas (Idem with references), whereas the proposal raised in this paper is consistent with these findings.

As mentioned above, room 86 has a special architectural element of great importance to the proposal raised in this article: the narrow opening 875 originally installed in the southeast corner of the room.[35] The opening is 82 cm wide. An opening of such a width in the outer wall of a site is at not all usual, and in my opinion, this opening is most suitable for use as a serving hatch for dishes to be passed from room 86 to the southern esplanade during gatherings there. It is worth noting that when reinforcement walls 801, 869 and 870 were built, their heights were designed exactly in accordance with the base of the hatch so as not to impair its function. Hence, the order of development of room 86 was as follows:

1. The room was erected in its original layout with the two pillars, connected to hall 77 and the southern esplanade by serving hatch 875.

2. Due to undermining, reinforcement walls 801, 869 and 870 were built below the height of the sill of the serving window, an indication that the back of the room remained in use.

3. Wall 873 was built, and the southern part of the room fell into disuse.

*4.3. The Southern Esplanade*

To the south of Qumran's complex of buildings lies a relatively flat natural plateau, covering an area of about 4000 m$^2$. Its shape is roughly triangular, ca. 60 m wide at the southern border of the complex of buildings and ca. 150 m long from north to south (Figure 1). The western side of the triangle is the edge of the slope leading down to Naḥal Qumran, and its eastern side is wall 951. From the beginning of Qumran's research, it was discerned that this low wall had no defensive function but was erected to separate the southern esplanade from the cemetery of the *yaḥad* community extending east from the esplanade, hence preventing defilement by the dead. However, why should an empty esplanade be protected from the impurity of the dead? As will be explained in detail below, in my opinion, it was in this esplanade that common meals were held during the annual convention of the sect.[36] The plan of wall 951, as it was discovered in the 1950s, was drawn up by De Vaux's expedition but was first published only about sixty years later by Humbert in 2006.[37] His investigation reveals very important details for the subject under discussion:

A. The wall indeed could not have played a defensive role and was never higher than one meter or slightly more.[38]

B. The fence is not a direct continuation of the eastern outer wall of the site (wall 900) and postdates it, as can be seen from the joint between them in L964. Later, Stacey (Stacey 2013, p. 29, Plan 4) concluded that wall 900 had previously turned west about 7 m further north and connected to wall 316 before hall 77 was built. This observation was confirmed

in the final publication of De Vaux's excavation by Humbert and Chambon (Humbert and Chambon 2016, pp. 159–60), who observed a kind of "elbow" (968) in wall 900 indicating two phases in the wall, which in Stacey's analysis is the turning point to the west.[39] Based on these findings, we could establish the successive phases of the eastern border in its southern part.

Phase 1: The eastern outer wall (900b) turned west at the "elbow" point (968) and surrounded the site to the south after the construction of the "triangle" but before the large structures (hall 77, room 86, *miqveh* 71 and reservoir 91) were built in the southern part of the site.

Phase 2: The site began to expand to the south. Hall 77 and room 86 were built, and a small *miqveh*, designated 71a in L187 by Humbert and Chambon, was constructed in the northern part of the area later occupied by *miqveh* 71 (Humbert and Chambon 2016, pp. 261–63). To surround the added structures, wall 900a was constructed as a continuation of wall 900b.

Phase 3: The large *miqveh* 71 was built. To enable its construction, the wall surrounding the site on the south was dismantled starting at connection point 964, and from this point southward, wall 951 was built to separate symbolically the entire area that had now been developed in the southern portion of the site from the cemetery area to the east. Wall 951 is probably the last structure built at the site before its destruction in 68 CE. Adding to this analysis, the findings of Magen and Peleg in this area result in an even more complete picture. In Phase 2, the use of the southern esplanade for burying pottery vessels with the remains of meals was discontinued, and they were buried in other areas. In the framework of Magen and Peleg's excavations, 18 excavation squares were opened in various places in the esplanade, and the unequivocal conclusion emerging from them is that the entire esplanade area remained permanently empty (Magen and Peleg 2018, p. 70). Why was such a convenient space never used for construction to expand the site to the south, and why was such an effort made to enclose it? What special importance did the southern esplanade have in the eyes of the people of Qumran to justify its symbolic protection by a wall? The facts that the wall is not defensive and is unlikely to have been used to repel wildlife[40] lead to the inevitable conclusion that the wall's function was to protect the southern esplanade from the impurity of the dead because many people gathered there to perform a particular act, and this act is related to the buildings added in the southern portion of the site. In addition, Pfann (2006, p. 160) is possibly right in remarking that the wall symbolically protecting the Qumran area from impurity included the caves in the marl step west of the site. As a matter of fact, in the case of caves 7Q, 8Q and 9Q, this inclusion was even physical, since the path leading to them passed within the enclosed area. In the excavations of Magen and Peleg, a burial field of pottery vessels was discovered in the northeastern part of the esplanade, some of them cooking pots filled with animal bones and date pits that had been buried there and then covered by a thin layer of hearth (Magen and Peleg 2018, pp. 58–59, 69). These vessel burials were dated to the first century BCE, and they are the oldest of the pottery burial fields discovered at Qumran. This use of the area stopped with the paving of the northernmost part of the esplanade adjacent to *miqveh* 71, hall 77 and room 86, since at its southeastern end, the pavement covers the burial field and seals it. From that time onward, the other burial areas of the site began to operate. This development clearly points to a constant increase in the number of people who needed to gather in the esplanade, starting from the core of the three buildings (77, 86 and 56–58) and continuing to the south. The paving is made of small stones, covers an area of 35.5 × 18.2 m (Magen and Peleg 2018, p. 47) and could accommodate several hundred people. The rest of the esplanade area remained as it was, since it was naturally flat and could accommodate many hundreds of diners.

### 4.4. Installations in the Southern Esplanade

In the northern part of the southern esplanade, the following facilities were installed (Figures 1, 2 and 5).

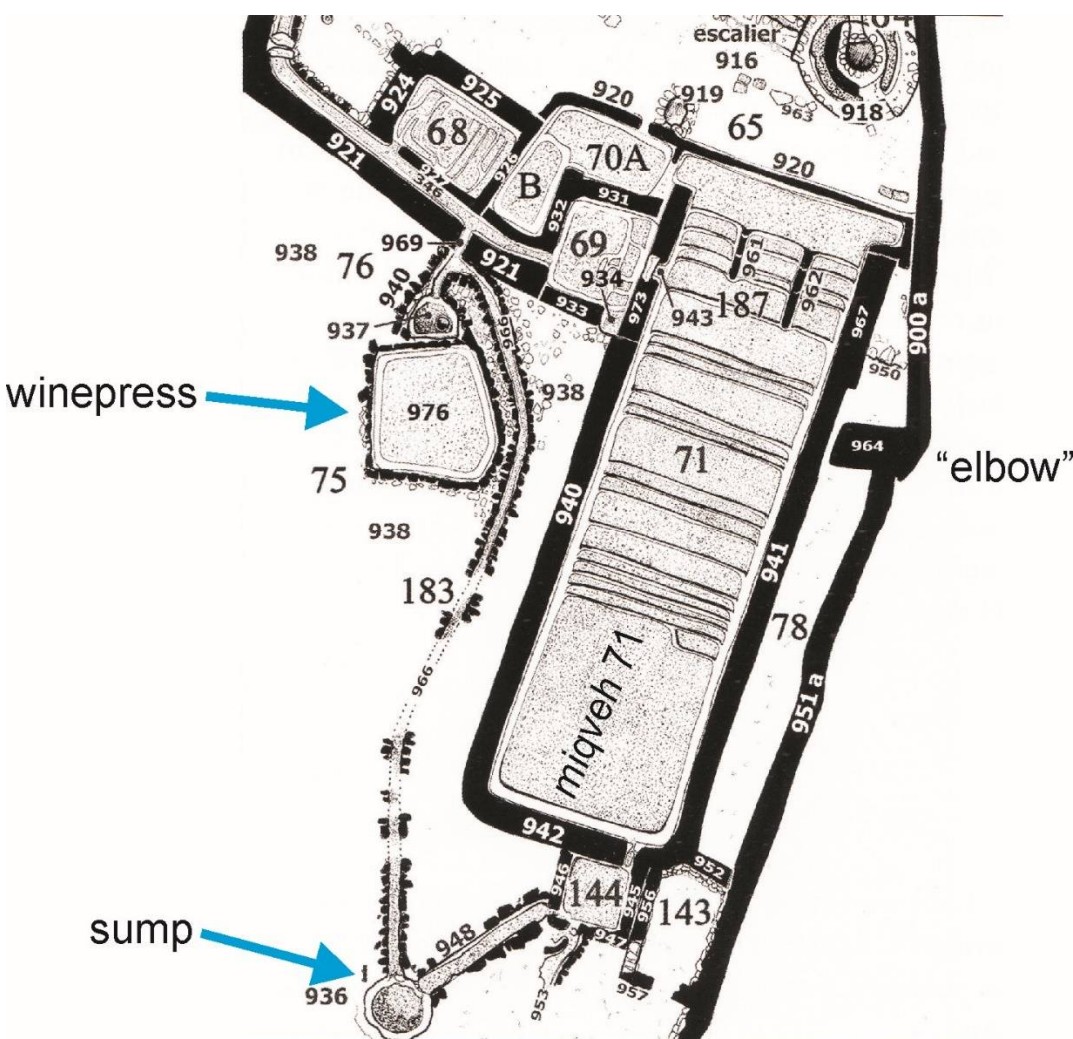

**Figure 5.** Based on Humbert and Chambon 2016.

### 4.4.1. Winepress (L75)

West of *miqveh* 71 is a large, plastered installation consisting of the rectangular surface 976 (about 3.5 × 3.5 m) and cistern 937 at its foot, into which liquids flowed from the rectangular surface. The cistern was abutted by a channel through which water could flow from water channel 346 via opening 969. De Vaux interpreted this complex as a facility of production of clay for the potter's workshop in the "triangle", but the typological examination of Humbert and Chambon has shown that there is no parallel to justify this installation's interpretation as a clay production facility and that it is better suited for treading grapes (Humbert and Chambon 2016, pp. 267–70).

### 4.4.2. Water Facilities

Water reached the esplanade area from a number of sources:

A. Channel 799 extended from the end of reservoir 91, crossing the outer wall of the site and passing into the esplanade area. The end of this channel was not preserved. Since the beginning of the channel was at the height of the reservoir's rim, a sweep was needed to enable water to flow through it (Galor 2003, p. 302; Humbert and Chambon 2016, p. 294, Planche X).

B. As mentioned above, a similar channel (944) extended to the south from the southeastern end of *miqveh* 71, but its continuation did not survive (Humbert and Chambon 2016, pp. 160, 238, 266, Planche IX).

C. From the starting point of the channel that directed water into the pit of the wine-press, channel 966 extended to the south, flanked the winepress on the east and poured water into the rounded sump 938, which could be used to purify the hands or utensils. Channel 948, which originated in room 144, also reached the sump. According to Humbert and Chambon, to the south of the paved area, the channels on the eastern side of the esplanade were connected in some way to the channels on the west side and could have been used for washing the paved area (Humbert and Chambon 2016, pp. 359–61, 365), perhaps similar to a facility installed in hall 77 for washing the floor.

### 4.4.3. Service Rooms

South of *miqveh* 71, two rooms were built that were open to the south toward the esplanade and were undoubtedly related to the activities held there. Among the pottery vessels found in room 143 was a complete bowl, on which the word מגע was written twice in extra-large letters. Lemaire (2003, p. 370) proposed to interpret it as a passive participle of the root נגע in the *hof'al* stem, meaning "defiled, impure". Animal bones and pottery fragments were also discovered in this room. Room 144 was used for activities related to the water that flowed into it from *miqveh* 71, and objects or food from Room 143 may have been purified there. It was built over the aforementioned channel 944 and from it led channel 948 toward the sump and channel 953 to the south (Humbert and Chambon 2016, pp. 272–73).

### 4.5. Miqveh 71

*Miqveh* 71 (Figure 5), which in my opinion was built for those gathered from all over the country for the covenant renewal ceremony, measures 19.1 × 4.9 m and, together with hall 77, is one of the two largest structures at Qumran. The *miqveh* was built of large stones and was entirely plastered (Galor 2003, pp. 302–4; Magen and Peleg 2018, p. 93; Humbert and Chambon 2016, pp. 259–66, Planche IX). Both its very large dimensions in relation to the estimated number of permanent residents at Qumran and its location at the far end of the site clearly indicate that it was not intended to serve the permanent residents, but it was most suitable for the immersion of the large crowds gathered in the southern esplanade south of rooms 77 and 86. The steps of the *miqveh* are not alike in their rise or their tread, in a manner somewhat reminiscent of the steps ascending from the Ophel to the Temple Mount in Jerusalem. In this latter case, some see it as an architectural technique aimed at making the ascenders and descenders slow down their steps and bow their heads instinctively (Ben-Dov 1982, p. 113). As in *miqveh* 56–58, the stairs are hewn to the entire width of the *miqveh* and occupy about two-thirds of the *miqveh*'s area, a feature allowing the movement of a large crowd but reducing the volume of water that could be stored. This is considered a characteristic feature of *miqva'ot*, distinguishing them from water reservoirs, in which a narrow staircase in one corner could be sufficient.[41] Channel 944 exits from the southeastern end of the *miqveh* toward the southern esplanade. Since the channel was at the height of the rim of the *miqveh*, water could flow through it only with the use of a sweep (Stacey 2013, p. 43; Galor 2003, p. 302). Similar to *miqveh* 56–58, the staircase of *miqveh* 71 is divided into three by two symbolic plaster partitions. This unusual division in these two *miqva'ot* cannot be the result of chance and is certainly an expression of a special concept on the part of the members of the sect. In my opinion, only full members of the sect were immersed in these two *miqva'ot*, and this division reflects an internal class division within the רבים that is quite possibly related to the three groups (priest, Levites and Israelites), into which the "מבקר of all the camps" divided the entire congregation (CD XIV, 5–6). The neophytes mentioned in this source apparently immersed together with the Israelites. In this context, it should be noted that in *miqveh* 43–48–49, which is not related to the complex discussed here, three plaster partitions divide the staircase into four (Galor 2003, pp. 300, 304, 311, 314). This singular arrangement may have been designed to address the need to separate candidates at different stages of their admission process.

The connection between *miqveh* 71 and the esplanade proposed here ostensibly conflicts with the site design shown in various maps of the site, where there is no access between the southern esplanade and the opening of the *miqveh*, but this difficulty is only apparent. All the walls around the "triangle" had openings that could not be restored today due to the poor state of preservation of the walls and the changes that took place after the destruction of the site in 68 CE, in Stratum III in De Vaux's terminology (Humbert and Chambon 2016, pp. 157, 59–61, "Le problème des accès aux annexes et à l'atelier de l'est"). Moreover, stratigraphic analysis of the series of pools and water facilities 69, 70, 68 and 72 along water channel 346, which ostensibly blocks access between the esplanade and the *miqveh*, shows that at no stage in the life of the site was this blockage complete. At each and every stage, the route was open (according to Humbert and Chambon 2016, pp. 158–161, 163, 267, 273; Galor 2003, p. 301; Stacey 2013), and the section of wall 921 separating them was built only in Stratum III, when the esplanade area became obsolete. During the year, the building unit consisting of hall 77, room 86 and *miqveh* 56–58 fulfilled the most important daily needs of the community gathering for study and communal meals after the required purification. This complex could easily have served a group of several dozen full members of the community who resided at Qumran, as well as its guests, while residents who were at various stages of candidacy immersed in the small *miqva'ot* on the site and dined separately. In the days of the annual convention leading up to the covenant renewal ceremony, when Qumran was flooded with members of the communities scattered throughout the country, the complex changed its appearance and adapted itself to the event. The crowds of guests gathered in the southern esplanade were seated on mats or carpets, the thousands of tableware vessels and serving utensils that had stood for a whole year arranged in piles in storage room 86 were passed out to the esplanade through serving window 875, and opening 876 was opened to connect hall 77 with the esplanade.

*4.6. Reservoir 91*

Water reservoir 91 (Figures 1 and 2), with a capacity of about 290 cubic meters, is the largest at Qumran. It was connected and combined with the two small reservoirs 83 and 85 (Galor 2003, pp. 295–296, 298, 310; Humbert and Chambon 2016, pp. 294–95, Planche X) and, according to some scholars, also served as a *miqveh* (Galor 2003, pp. 310, 16–17; Magness 2002, p. 158). Similar to *miqveh* 71, channel 799 exits from the southern end of reservoir 91 toward the southern esplanade at the height of the rim of the reservoir. Water could flow through the channel toward the esplanade only with the use of a sweep (Stacey 2013, pp. 42–43; Galor 2003, p. 298). The channel crosses wall 786 and enters the esplanade. According to Stacey, water reservoir 91 was built in the last stage of the development of the site in the days of Herod, after *miqveh* 71 and buildings 77 and 86 were already in use, and it was added to them (Stacey 2013, pp. 32, 42, 50).

**5. Discussion and Conclusions**

According to most scholars, at the end of the Hasmonean period (167–37 BCE), the southern boundary of the Qumran site was wall 316. At the end of this period, or in the early days of Herod, the buildings of the "triangle" were added on the eastern border of the site and were surrounded by wall 900, which turned sharply to the northwest and connected with the complex. The entire area south of these walls remained empty, and its only use was to bury pottery vessels with remains of meals. During the reign of Herod the Great (37–4 BCE), a turning point occurred, and impressive construction works were carried out in the southern part of Qumran that changed the face of the site. The burial of the remains of the meals in the southern esplanade was stopped, part of the wall surrounding the "triangle" in the south was dismantled, and a large assembly hall was built with a storage room for tableware and serving utensils. A large *miqveh* and a large reservoir were also built, from both of which water channels led to the esplanade, the entire esplanade was separated from the cemetery by a low wall, and its northern part was paved. Stacey (Stacey 2013) makes a careful distinction between five different stratigraphic stages

in Herod's days, a division that differs from that of his predecessors, but there is general agreement that all these significant changes in the southern part of the site, or almost all of them, took place in Herod's days. To this impressive development we must add an equally impressive fact: no residential buildings were constructed anywhere on the site in parallel with the construction of the new public buildings to the south. The combination of these two facts should lead us to three conclusions:

A. These buildings were not built for the permanent residents of Qumran but rather for people who came to the site every year for a very short period of time and therefore did not need housing.

B. It is probable that these visitors also co-financed the construction works carried out in a relatively short period of time during Herod's reign, and it is unlikely that the small group that resided permanently at Qumran was able to finance the works and the logistical needs of the annual gatherings. It is likely, therefore, that the impressive coin hoards discovered at Qumran, comprising more than 1200 coins, most of them Tyrian shekels,[42] were brought there by members of the *yaḥad* communities throughout the country.

C. The empty southern esplanade actually became the most important place on the site, with the impressive buildings at its northern edge intended to serve and satisfy the religious and material needs of the crowd that gathered there every year.

The founding community of the *yaḥad* sect settled at Qumran at some time in the first half of the first century BCE on the ruins of a biblical site that had stood desolate for more than five centuries.[43] Over time, the movement developed. Groups of Essenes lived near the water sources along the shores of the Dead Sea, and other communities arose throughout the country, some within localities alongside other Jews and others in separate small localities that they established for themselves. It seems that the diversity of the communities that made up the sect, reflected both in the classical sources and in the scrolls as presented here, was a consequence, at least in part, of the absorption in the sect of structured communities in different parts of the country who practiced a different way of life from that of the main currents in Judaism at that time and held a worldview similar to that of the Qumranites. This diversity of communities is reflected in the contradictions in the various sources, such as the sharing of property according to Josephus, Philo of Alexandria and the Community Rule, as opposed to the obligation of the member to set aside money from his private funds for community needs according to the Damascus Document. Out of this reality was born the halakhic duty of the members of all the groups that made up the sect to come to Qumran once a year and participate in the covenant renewal ceremony. Some of the ten copies of the Damascus Document that were discovered in the Qumran caves date from the end of the Hasmonean period or the beginning of the Herodian period (see Ada Yardeni and Emanuel Tov in Baumgarten 1996, pp. 1–2), which is to say they were written in the period in which the rule governing the relations of the communities that made up the sect was actually formulated and are hence remnants of its first copies. This dating coincides with the beginning of the expansion of the Qumran site to the south, the transformation of the southern esplanade into the focus of the site and the building in its northern part of the structures necessary for the annual national gathering during Herod's reign. Moreover, the earliest of these copies, 4Q266, is so replete with corrections and deletions that it appears to be a sort of draft or personal copy made during the process of crystallization of the wording of the rule (Ibid.).

The archaeological complex that emerged at the end of this process is unique and has no equal in the archaeology of Israel: a compound of 4000 m$^2$ symbolically separated from the cemetery where more than 2000 people could sit on mats after immersion in one of the largest *miqva'ot* in the country at the time, fed one loaf of bread and one bowl of stew per person and watered with must and water as described for the sect's common meal (see Josephus *J.W.* 2.8.5; Philo of Alexandria (Eusebius of Caesarea, *Praeparatio evangelica* 8.11.11); 1Qs VI, 3–5). The members of the *yaḥad* sect were modest people, and contentment with little was a cornerstone of their worldview. The hundreds of adult men who stayed for a few days each year at Qumran were not regular road station visitors of the time and

would not be expected to leave the same archaeological footprint. These were humble and ascetic people who congregated in days of spiritual awakening. The younger ones among them could spend the night outdoors without leaving any archaeological traces. In this regard, it is important to take into account that the gathering was held during the Shavuot period at some time in May, when there was no chance of rain in the Qumran region and the temperatures were a maximum of 35 °C during the day and a minimum of 22 °C at night.[44] In such conditions, there was no need to build special infrastructure to accommodate most of the participants, who were exclusively adult males who came for a few days each year.

For the older ones, caves were prepared, and sheds were put up on surfaces that were level or especially leveled for this purpose. The guests in the caves, due to their character and the special circumstances of the visit, had few material needs and therefore left almost nothing in about one hundred fifty years of use of the caves. During the year, some of these caves were probably also used for other purposes, such as storage or deposits. Even caves that were defined by researchers as unfit for human habitation could serve as accommodations for a short period of time.[45]

The archaeological finds from both the natural limestone cliff caves and those carved in the marl terrace surrounding Qumran have presented some unsolved challenges to researchers from De Vaux onward (see Mizzi 2017b, especially 144–145). On the one hand, they are in line chronologically and typologically with the finds made at the site of Qumran. On the other hand, their cumulative amount is small for over a century of use. Over time, various researchers have raised, among others, the suggestion that the caves were used for residence for very short periods of time each year. Hence, Humbert (1994, p. 176) spoke of temporary residence in the caves, while Mizzi considered, among other options, the possibility that the caves experienced "temporary, intermittent occupations" (Mizzi 2017b, p. 149) or "functioned as mere sleeping quarters", (Mizzi 2017b, p. 151) and in Hirschfeld's opinion, the caves served only as temporary living quarters.[46]

In Crawford's opinion, the limestone cliff caves could not "have been used for long-term habitation, any more than a few days of weeks [a year]", and "the marl terrace caves were not used for permanent, full-time habitation, although they seem to have been used as temporary sleeping quarters or work spaces", as proposed by Mizzi (Crawford 2019, pp. 117, 131, 135–136, 215, 310). Interestingly, she also concludes that "it is possible that temporary tents or huts were used, but this would only be on 'overflow' occasions" (Op. cit. p. 215, note 195), a view that fits the proposal advanced in this paper.

Fresh and very important information in this regard is currently coming from a comprehensive geological and archaeological survey of the caves conducted by the Israel Antiquities Authority.[47] Among their conclusions are the following: (1) the number of artificial caves carved in the marl is greater than previously thought and is at least 29, connected by a network of paths in what the authors call a "cave village" (Op. cit. pp. 92–93); (2) there were more smoothed stone surfaces than those found by Broshi and Eshel, leading the authors to accept the possibility raised by Broshi and Eshel (and earlier proposed by de Vaux) of the use of tents and sheds in the "northern plain" (Op. cit. p. 90); and (3) the findings in the caves, including the ones newly discovered, link them to the site of Qumran and present the same picture of meager quantity, pointing to noncontinuous occupation and leading the authors to favor the interpretation that the caves were inhabited only for short periods of time each year (Op. cit. pp. 94–96).

The fit between this archaeological complex and the descriptions of the annual gathering in the scriptures cannot, in my opinion, be the result of chance. It is possible that, between the initial crystallization of the rules and the construction of the southern compound in the days of Herod and the destruction of the site in the days of the Great Revolt against the Romans, there were developments and changes in the structure of the sect and, as a result, changes in the rules themselves, reflected in the differences between the Damascus Document and the Community Rule. We cannot restore them all.

It seems that the establishment of the annual gathering as a major event uniting all the communities remained a cornerstone until the destruction of the site and the elimination of the sect. The proposal presented here may also shed light on the possibility that the hoards of coins discovered at Qumran are related to the donation of a half-shekel by members of the sect. Regarding the half-shekel tax from the perception of the people of Qumran, it is explicitly stated in scroll 4Q159 that each Jew should pay it only once in his lifetime: "only one [time] shall he give it all his days" (רק [פעם] אחת יתננו כול ימיו). Liver (1963, pp. 190–98) correctly pointed out that this statement, based on the simple understanding of the biblical passage in which a half-shekel was collected only at a census as a one-time donation (Exod 30, 38), is an expression of opposition to the Pharisaic law that made the once-in-a-lifetime donation an annual one, articulated only after the separation of the *yaḥad* sect and its non-recognition of the Temple and its priests. Liver also correctly shows the connection of the donation of a half-shekel with the texts in the Community Rule and in the Damascus Document (Liver 1963, pp. 196–97) which, in my opinion, describes the annual gathering of the sect. The occasion on which the member of the sect had to donate the half-shekel was not stated in the Qumran scrolls, but there is information about the two circumstances in which a census of all the members was conducted, and since the half-shekel was donated during a census, the time of the census may also include information about the donation of the half-shekel:

1. According to the Community Rule (1QS V, 20–25), from the day of each member's full membership in the sect, in the census held every year, his hierarchical position within his community is redetermined "according to his intellect and deeds", and this is recorded in an updated hierarchical list.

2. As aforesaid, another census is mentioned in the Damascus Document XIV in the framework of the "assembly of all the camps", in which the main task of the "מבקר of all the camps" is to conduct a hierarchical countrywide census based on the updated lists of each community.

In my opinion, this evidence should lead us to the understanding that the once-in-a-lifetime payment of a half-shekel of any member of the *yaḥad* was made when he first participated in the covenant renewal ceremony, which was the official seal of his joining the sect after successfully completing his candidacy within his community.[48] The new entrants who came to Qumran to attend the covenant renewal ceremony for the first time brought with them Tyrian shekels as the half-shekel donated to the Temple in Jerusalem at that time by mainstream Jews. This may explain the impressive accumulation of hundreds of Tyrian shekels, alongside other coins, in the Qumran hoards.[49]

Another unique find at Qumran, for which no satisfactory explanation has been offered so far, is the burial of cooking pots containing animal bones, apparently remains of meals, throughout the life of the site.[50] The reason for the difficulty of explaining this particular phenomenon, for which there is no parallel, is that it cannot be linked to any written testimony, whether in the normative writings of Judaism or in the writings of the *yaḥad* sect. In light of what has been said here, it seems more likely that these are remnants of communal meals at the annual conventions rather than of meals on ordinary days.

The question of the number of permanent residents of Qumran and where they dwelled has concerned researchers ever since the site was discovered. Since most of the site is occupied by public facilities such as the *miqva'ot*, reservoirs, dining room, kitchen and writing room, researchers who estimate that between 150 and 200 people lived at Qumran assume that these people lived mainly on the second floors of buildings, which have not survived, and in some of the marl caves near the site. Other researchers believe that according to the available data, only 10–15 people (Humbert 1994, p. 176), or at most about 50 people,[51] lived permanently at Qumran. In light of all the excavations and surveys conducted in the last 70 years, a picture emerges of a community of some dozens of people, but probably less than a hundred, whose members lived on the site in caves and perhaps small sites nearby, alongside an infrastructure suitable for hosting hundreds or even thousands of people for a short time year by year. Of course, this does not mean that individual

members or even groups did not visit Qumran during the year, but these visits were not obligatory and do not compare with the obligatory annual general meeting. This infrastructure consisted of all the public facilities at Qumran, which were appropriate in nature to the special lifestyle of the members of the *yaḥad* sect with its needs for purification, eating together and studying and discussing sacred texts, together with an array of dozens of caves and makeshift sheds that accommodated some members of the congregation who came from afar. Among these public buildings, the *miqva'ot*, which in their size and number in relation to the size of the site are unique in the country, stand out (Galor 2003, pp. 291–293, 313, 317).

As I have tried to show, in my opinion, not only was Qumran used for the absorption of all the members of the sect throughout the country in its obligatory annual gathering, but the annual gathering was the main reason for establishing the infrastructure. Unlike the gatherings around pilgrimage centers, monasteries or tombs recognized in many cultures all over the world and throughout the ages, the annual gathering at Shavuot of all the *yaḥad* communities was obligatory and a condition for continued membership in the sect. Hence, the establishment of the sect's center was essential. Some dozens of permanent residents of Qumran, perhaps with the help of small Essene groups living nearby close to the springs on the shores of the Dead Sea, had to host many hundreds of people at the site once a year in ever-increasing numbers. The site of Qumran, with its facilities, caves and surfaces, accords with the evidence for the annual gathering that emerges from the scrolls. No other known site is suitable for such a purpose. Our proposal, therefore, is in line with the view of most scholars that Qumran was the main site of the *yaḥad* sect, the caves in its vicinity and the scrolls discovered in them are related to it, and the *yaḥad* people mentioned in the scrolls are related to the Essenes mentioned in the literary sources.[52]

**Funding:** This research received no external funding.

**Institutional Review Board Statement:** Not applicable.

**Informed Consent Statement:** Not applicable.

**Data Availability Statement:** Not applicable.

**Conflicts of Interest:** The author declares no conflict of interest.

## Notes

1    In this paper, I use the terms "*yaḥad* sect", "*yaḥad*" or "the sect" to refer to all of the communities reflected in Qumranic scrolls, in which these communities are termed *yaḥad*, as is common in the Community Rule, or עדה, as is common in the Damascus Document and related texts. This accords with the conception espoused by many scholars (see Collins 2010, pp. 65–79; Hempel 2011; Schofield 2009, especially chapters 3 and 5; Metso 2009; Mizzi 2017a), as well as by myself, that all of those communities were linked in a countrywide organization and that their members were Essenes, who are recorded by Josephus, Pliny the Elder, Philo of Alexandria and other classical authors, and that Qumran was a major center of this network of communities.

2    Although, in Falk's opinion, some type of renewal of covenant ceremony was held among other Jewish groups as well (Falk 1998, pp. 225–26).

3    The verb also recalls the passing of the torch between the halves in the Covenant of the Pieces (Gen 15:17).

4    See the unique combination of the different meanings of the verb in a single verse in Jer 34:18: "the people that violated my covenant" (הָאֲנָשִׁים הָעֹבְרִים אֶת-בְּרִתִי) together with "and passed between its halves (of the calf)" (וַיַּעַבְרוּ בֵּין בְּתָרָיו), evoking the Covenant of the Pieces.

5    By "Community Rule" and "Damascus Document", I mean the families of manuscripts preserving these traditions (see Crawford 2019, p. 224, note 18), although, for convenience, I use the reference numbers of 1QS for the former and those of the Cairo Damascus copy for the latter. On the connection between the two compositions and the communities reflected in them, see Hempel 2009 with earlier references; Hempel 2010, especially pp. 130–31 with earlier references; Collins 2010, pp. 5–6, 54–56. In Falk's opinion, the common source that served both compositions cannot be later than the end of the second century BCE (Falk 1998, p. 228).

6    According to Newsom, this symbolizes the entrance of the new member into the sect (Newsom 2004, p. 118).

7    (Hempel 2010, pp. 127–29; Newsom 2004, pp. 117–27). On the relative dating of the different manuscripts, see (Collins 2010, pp. 52–53; Collins 2011, pp. 12, 15; Grossman 2016, especially pp. 320–25).

8    (Milik 1962, pp. 181–83). On this point, see especially (Metso 2004, pp. 328–30).

9    (Alexander and Vermès 1998, pp. 209–16). "Walking the path" in the text is symbolic, and it is not known whether the reality in which the sect members walked the paths that connected the caves in the area to the gathering place influenced the author of the scroll.

10   On the possibility that the term מבקר is recorded in an ostracon found in Qumran, see (Puech 2007).

11   (Fraade 2011, p. 232). In his opinion, its influence is reflected in 4QMMT (Fraade 2003). On the liturgy in the ceremony, see (Falk 1998, pp. 219–36). In his opinion (p. 236), because of its centrality, "the influence of this central, annual ritual pervaded all liturgical life for this group".

12   Reconstructed after 4Q266 11; 4Q270 7 II; 4Q269 16, lines 15–18. See (Wacholder 2007, pp. 106–7).

13   (Baumgarten 1996, p. 78). In Fraade's opinion, it was implemented "on or just prior to the festival of Shavu'ot": (Fraade 2003, pp. 155–56).

14   4Q270 7 II; (Baumgarten et al. 2006, p. 156, especially 4Q267 11, 16–18; Baumgarten 1996, pp. 76–77; Hempel 2000, p. 80).

15   On the varied origin of some of the Qumranic scrolls, see (Anava et al. 2020, especially pp. 1221–22, 1226–28).

16   See, for example, (Baumgarten and Schwartz 1994, pp. 53, 57), who translated "settlement"; (Wacholder 2007), who sometimes translated "habitation" (pp. 13, 57, 93, etc.) and sometimes correctly translated "session" (101, 327, 354, etc.). On p. 97, the word מושב is translated "habitation" for CD 13:20 and "session" for CD 14:3.

17   The term is used very similarly in Ben Sira to express the specific hierarchic seating place of a person (Sir 7:4; 12:12).

18   This is also preserved partially in 4Q267 9 V.

19   This is also preserved partially in Qumran scrolls; see (Baumgarten 1996, pp. 109–10).

20   See proposals for reconstruction of the very damaged text in (Broshi 1992, p. 37; and Baumgarten and Schwartz 1994, pp. 56–57).

21   (Naudé and Miller-Naudé 2015, pp. 93–97). See note 14 on pp. 93–94 with numerous examples, including the two phrases discussed here. The few exceptions to this rule are in the Temple Scroll, expressing measurements (pp. 96–97). On the contrary, see on pp. 100–102 the syntactical ways to express "each" and "every". Muraoka (2020, p. 133) agrees with them; see his translation "the inspector over all the camps" in note 4 on p. 150.

22   See (Metso 2002, pp. 440–41) with earlier references.

23   One cannot rule out the possibility that the pluralization of the title רב as רבנים instead of רבים in mainstream Jewish circles was adopted as a sort of disambiguation from what were seen in their eyes as marginal circles.

24   "and why did he use to call them רבים? Because they were great (רבים) in the Torah" (Pesiq. Rab Kah., *ki tiśśaʾ*, Piska 1); "רבים. These are Doeg and Ahithophel, who were great (רבים) in the Torah" (Tanḥ., *ki tiśśaʾ*, Piska 4); "and also if you sit among רבים" (Sir 31:22). This last example in Ben Sira is very similar to descriptions of the sitting of the רבים in Qumran scrolls.

25   See (Licht 1965, pp. 115–16; Metso 2002, pp. 439–40) with earlier references.

26   It is possible that החבר (CD XIV, 16) is an abbreviation for חבור ישראל (CD XII, 8), an epithet for the entire *yaḥad* sect: (Baumgarten and Schwartz 1994, pp. 50–51; Wacholder 2007, p. 350).

27   (Qimron 1990; Wacholder 2007, pp. 78–79, 306–308, especially 308). The lack of the beginning of the phrase has engendered other interpretations. For a summary of them see (Wacholder 2007, pp. 306–8).

28   Unless otherwise stated, all the numbers of loci and finds in this paper are according to (Humbert and Chambon 2016). Figure 1 is based on (Humbert 1994, p. 204) and (Humbert and Chambon 2016). The other figures are based on (Humbert and Chambon 2016). I would like to thank the archaeologist Evgeny Aharonovich for his kind help examining and measuring the findings.

29   Late first century BCE through the early first century CE. See *Dio Chrysostom, With an English Translation by H. Lamar Crosby*, LCL, vol. V (William Heinemann and Harvard University Press, London and Cambridge, 1951), pp. 378–379. Dio Chrysostom's words are quoted by Synesius (late fourth century through the early fifth century CE). His testimony does not depend at all on that of Pliny (Taylor 2009, pp. 6–7). He reports on an Essenian "city" "somewhere near Sodom" (κειμένην αὐτά που τὰ Σόδομα). On the conformity of this report with Khirbet Qumran, see (Taylor 2010).

30   Although in Stacey's opinion, hall 77 was constructed in the stage called by him "Herod III", at which time L58, the eastern part of the *miqveh* (and possibly even L56, its western part), had fallen into disuse, and *miqveh* 71 was built (Stacey 2013, pp. 30, 39). According to his reconstruction, room 86 was built in the following stage, "Herod IV".

31   There was very possibly another dining room, which could have served the candidates in the northwestern part of the site (Magness 2004, pp. 102–3).

32   (Humbert and Chambon 2016, pp. 71–75; Wagemakers and Taylor 2011). In Wagemakers's and Taylor's view, the pillars in hall 77 probably supported wooden or cloth partitions to separate between men and women or people of different hierarchical status. In the opinion of Humbert and Chambon, they served for offerings of first fruits.

33   In De Vaux's opinion, the cause of the undermining of the room was the earthquake that occurred in 31 BCE, and his opinion was accepted by Magness (Magness 2004, p. 92) and others (Wagemakers and Taylor 2011, p. 135). In Stacey's opinion, the room was undermined much later by the quarrying of the large water reservoir L91 (Stacey 2013, p. 50). Humbert and Chambon

34    (Humbert and Chambon 2016, pp. 329, 333–37), however, think that the hoard of vessels was damaged not by a shock to the structure but by a fire very shortly before the final destruction of the site in 68 CE.

34    See extensive detail in (Pfann 2006, p. 163).

35    See (Humbert and Chambon 2016, pp. 329, 333), especially picture 135 on page 326 and Figure 136 on page 327.

36    For a summary of opinions about the purpose of the wall, see (Branham 2006, pp. 117–31).

37    (Humbert 2006, pp. 20–22; Humbert and Chambon 2016, pp. 39–41, Planche XII).

38    In an initial publication, Humbert estimated its maximum height at 1.4 m (Humbert 2006, p. 22). In the final publication of the site in 2016, the maximum height was estimated at only 1 m (Humbert and Chambon 2016, p. 39).

39    Humbert and Chambon call the older part of the wall (north of the "elbow") 900b and the later part (south of the "elbow") 900a. In Figures 60 and 61, an error occurred, and both parts were called 900a.

40    According to (Magen and Peleg 2018, pp. 129–30). Because of its low height and non-continuous nature, it could not have prevented the intrusion of animals, and in any case, why would the residents of Qumran want to prevent animals from entering an empty space outside the built-up area?

41    See (Galor 2003, p. 305) with earlier bibliography.

42    See the summary of coin finds in (Magness 2002, pp. 188–209). On the possibility that the Tyrian shekels were raised as a substitute for the half-shekel tax in its biblical form, that is, once in a lifetime, see (Liver 1963).

43    See (Mizzi 2017a, pp. 14–17). For the once proposed and now mostly rejected connection of the beginning of the sectarian settlement in Qumran with the Teacher of Righteousness, see (Collins 2020, especially p. 169).

44    I thank the Israel Meteorological Service for this information and their kind help.

45    In the archaeological research the few finds made in the caves, despite the long period of their use, have already been interpreted by Humbert (Humbert 1994, p. 176) as representing temporary residence in the caves. Broshi and Eshel interpreted smoothed stone surfaces as bases of tents or sheds, although not for guests but for the permanent residents of Qumran during heavy rain, for Sabbath stays, or for the absorption of refugees during the Great Revolt (Broshi and Eshel 1999, p. 339). In the opinion of Magness (2002, p. 70), most of the permanent residents of Qumran lived in tents and sheds around the settlement. For a summary of opinions, see (Collins 2010, pp. 180–82).

46    (Hirschfeld 2004, p. 43). In his opinion they were used by laborers, shepherds, hermits and mere passersby.

47    The findings will be published in O. Sion, A. Ganor, E. Klein, H. Hamer, and O. Amihai, Survey and Excavations Project in the Judean Desert. See a preliminary report in (Cohen et al. 2021).

48    This option was considered but not accepted by Liver, as well as Magness (2002, pp. 190–93).

49    See also (Magness 2002, pp. 188–93) with rich bibliography on the numismatic findings.

50    See the summaries of this findings in (Humbert and Chambon 2016, pp. 39–41, 59–64; and Magen and Peleg 2018, pp. 127–30).

51    Patrich (Patrich 2000, p. 726–27) considers the possibility that Essenes from Jerusalem and other places may have come to Qumran for holidays and celebrations.

52    The fact that the annual gathering of all the *yaḥad* communities was performed at Shavuot (the Feast of First Fruits) raises the question of whether the members of the groups who came from all over the country fulfilled the commandment to bring their first fruits by bringing them to Qumran instead of the Temple in Jerusalem, and whether is this reflected in archaeological findings, as proposed by scholars such as Humbert (1994, p. 201; Humbert 2006, p. 36) and Milik 1957, p. 77). This question is beyond the scope of this article, especially in light of the fact that according to the halakha of the *yaḥad* it was forbidden to eat bread from the new harvest before the first fruits of the wheat were brought to the Temple: 4Q251 9. See (Baumgarten et al. 1999, pp. 34–35).

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
