# Peer review of "The Covenant Renewal Ceremony as the Main Function of Qumran"

_religions, doi:10.3390/rel12080578_

Round 1

Reviewer 1 Report

a) The word "halkakhic" is rabbinical, and should not be used for the Qumran material.

b) The carefully buried bones suggest a kind of Passover leftover.

Author Response

Dear reviewer,

Thank you very much for your review.

In the use of “halakhic” for the different Jewish currents, not only rabbinic, I follow the very  usual practice of many researches like J.M. Baumgarten et al., Qumran Cave 4. XXV. Halakhic Texts [DJD 35], Oxford, L. Schiffman, The Halakhah at Qumran, Leiden 1975, or A. Shemesh and C. Werman, “Halakha at Qumran: Genre and Authority”, DSD 10 (2003), etc. Even terms as “sectarian halakhot” became very usual in the research of Qumran.

The carefully buried bones at Qumran constitute an enigma. You are right that they can suggest a kind of Passover leftover, but in my opinion there was only one time in the year, in Shavuot, for a special gathering and meals of all the members of the sect. Anyway, the leftovers of the Passover offerings must be burned (Exodus 12:10), not buried.

Many thanks

Reviewer 2 Report

Coming from another academic discipline, I would have liked to see a stronger introduction with a discussion of methods and research questions, but that might a different academic tradition.

Personally, I would have been interested in the role of gender played in the events and settings discussed in the paper.

Author Response

Dear reviewer,

Thank you very much for your review.

Please pay attention, that the introduction in the paper is not limited to the “introduction” section, but in fact each chapter or section of the paper begins with an introduction of the particular issue discussed in it.

In the annual gathering discussed in the paper only men participated.

Many thanks.